# Impact of Environmental and Health Risks on Rural Households’ Sustainable Livelihoods: Evidence from China

**DOI:** 10.3390/ijerph182010955

**Published:** 2021-10-18

**Authors:** Wei Wang, Chongmei Zhang, Yan Guo, Dingde Xu

**Affiliations:** 1Department of Agriculture and Forestry Economics and Management, College of Management, Sichuan Agricultural University, Chengdu 130062, China; wangwei@sicau.edu.cn (W.W.); 2020209054@stu.sicau.edu.cn (C.Z.); 2Department of Data Science, College of Information Engineering, Sichuan Agricultural University, Chengdu 130062, China; guoyan@sicau.edu.cn; 3Sichuan Center for Rural Development Research, College of Management, Sichuan Agricultural University, Chengdu 130062, China

**Keywords:** risk impact, peasant households, sustainable livelihood

## Abstract

China has entered a “post-poverty alleviation” era, where the achievement of sustainable livelihoods by farmers has become a focus. This study used the China Family Panel Studies (CFPS) database, which was constructed based on an analysis of the DFID sustainable livelihood framework, and built a sustainable livelihood index system for farmers using the entropy weight method to measure the weights of sustainable livelihood indexes and calculate a sustainable livelihood index. This study used the Tobit model to discuss the impacts of different types of risk on the achievement of a sustainable livelihood by farmers. The results showed that environmental risk, chronic disease risk, and major disease risk all had significant negative impacts on the ability of farmers to achieve a sustainable livelihood. The impacts of major disease and chronic disease risks on the achievement of a sustainable livelihood by farmers living in plain areas were stronger than those associated with environmental risk. In China, the environmental risks were complex and diverse and were the most important factors that affect the achievement of a sustainable livelihood by rural households in mountainous areas. Chronic disease risk was also an important adverse factor that affected the achievement of a sustainable livelihood by rural households in mountainous areas.

## 1. Introduction

China achieved a comprehensive victory regarding poverty alleviation and accomplished the arduous task of eliminating absolute poverty. All 98.99 million rural residents living below the current poverty line were lifted out of poverty. This included residents of 832 poverty-stricken counties and 128,000 poverty-stricken villages [1,2,3]. Since then, China has entered a later stage of poverty alleviation. Optimization of the direction of this initiative and implementation of poverty alleviation policies will require a transition from solving absolute poverty to alleviating relative poverty, as well as a transfer from phased poverty reduction to sustainable poverty reduction. Achievement of a sustainable livelihood for peasant households has become a focus of the “post-poverty alleviation” period. In addition, China is currently in a period of accelerated social transformation; therefore, achieving a sustainable livelihood by peasant households is urgent.

The impact of risk shock on the livelihoods of farmers has long been a focus of domestic and foreign academic circles. Risk shock refers to an unexpected situation with a certain impact on farmers regarding daily production and living processes, which leads to a decrease in income or an increase in expenditure. It may even cause difficulties in family life. Due to their strong vulnerability and relative lack of livelihood capital, farmers are frequently faced with risk shocks. Most farmers lack the required risk management processes. Once they are affected, significant impacts on production and life occur, affecting their ability to achieve a sustainable livelihood in the long run.

This study analyzed a sustainable livelihood framework and the China Family Panel Studies (CFPS) database and built an index system to aid with achieving sustainable livelihoods for farmers. Using this, the weights of sustainable livelihood target indexes were measured with the entropy weight method and the sustainable livelihood index of farmers was measured. The Tobit model was used to discuss the impacts of different types of risk on the achievement of a sustainable livelihood by farmers. We aimed to explain the negative impact of risk shock on the achievement of a sustainable livelihood by farmers. This was discussed at the microcosmic level from the perspective of farmers from different regions and according to different agricultural risk categories. We developed an effective risk prevention and response strategy and decision-making ideas for farmers to increase the number of farmers that can achieve sustainable livelihoods. Our results have practical reference value.

The rest of this paper is arranged as follows: The second part presents a literature review and relevant theoretical analysis. The third part introduces the sources of data that were used in this study, the selection and weight measurement methods that were used to assess sustainable livelihood indicators, and the variables and models that were adopted. The fourth part is the main part of the paper; it presents a correlation analysis of the impacts of environmental and health risks on the achievement of a sustainable livelihood by farmers, as well as the results of the endogenous problem test and robustness test. The fifth part presents a discussion of the results, and the sixth part presents the conclusion and relevant policy suggestions.

## 2. Literature Review and Theoretical Analysis

### 2.1. Literature Review

#### 2.1.1. Research on the Impact of Risk

Risk shocks have been studied around the world for a long time. Scholars believe that risk shocks are an important manifestation of farmers’ vulnerability and uncertainty [4,5,6]. As farmers are often vulnerable, risk shocks will impact them significantly [7]. Domestic scholars also studied risks in the poverty research field [8]. Previous studies on risk shocks generally included risk shock categories, the impacts of risk shocks, attitudes to risk, and risk responses [9,10,11,12,13,14,15].

Research results on the impact of risk are abundant, and the risks faced by farmers are characterized by diversity and correlations [16]. According to previous research results, risks can be divided into natural risks, market risks, policy and institutional risks, education risks, health risks, and employment risks [17,18,19]. Some scholars have integrated the viewpoints of domestic and foreign scholars to show that agricultural risks can be divided into six main categories: international risk, institutional risk, market risk, technical risk, food security risk, and natural risk [20]. Some studies focused on the risk impact of agricultural production. For example, the Risk Management Service under the United States Department of Agriculture divided agricultural risks into output risk, price or market risk, policy risk, human risk, and financial risk, and stated that these risks have important impacts on agricultural production and operation activities. Scholars concluded that the five most common risk types in agricultural production are agricultural production risk, market risk, institutional risk, personal risk, and financial risk [21].

There were many academic studies on the adverse effects of risk shocks. Scholars analyzed the impacts of risk shocks from the perspective of livelihood capital [22,23,24]. For example, a quantitative analysis model was developed to quantify the impact of livelihood risks on farmers’ livelihood capital through a semi-structured survey of farmers from the Shiyang River Basin of Gansu Province, China [25]. Natural disaster risk shock has adverse impacts on agriculture, the economy, and land use [26,27,28]. Drought shock is the key factor that causes rural households to become poor [29]. Some scholars also measured and compared the losses of farmers that were affected by natural risks to agricultural production, such as drought, heat, and hail [30]. Market risk also causes a loss of income for farmers [31,32]. Scholars used micro household data from the 2013 China Household Finance Survey (CHFS) to analyze the debt risk level of urban households in China and concluded that, compared with the impact of unemployment, a downward fluctuation in housing prices has a greater negative impact on household debt risk [33]. As for the impact of policy risks, scholars believe that risks faced by ecological migrants have doubled the risks caused by the relocation shocks that are faced by farmers [34]. In addition, an empirical research analysis showed that health risk shocks negatively impact the peasant household economy through two channel mechanisms: the crowding effect and the emotional effect [35].

#### 2.1.2. Research on Sustainable Livelihoods

The concept of a “sustainable livelihood” was derived from creative research on poverty by Sen Conway, and Chambers. It refers to the assets, capabilities, and livelihood activities that people possess and acquire and can be used to seek out and improve long-term living conditions. In the process of studying poverty, scholars have paid more attention to the deeper causes of poverty, such as poverty in opportunities and the restrictive environment of livelihood development [36,37,38]. This concept was widely accepted by scholars. In 1992, the United Nations Conference on Environment and Development (UNCED) introduced the concept of sustainable livelihood and pointed out that the achievement of stable livelihoods could enable the coordinated development of relevant policies, the elimination of poverty, and the sustainable use of resources. The Copenhagen Declaration, adopted in 1995, stated that “To make full employment a priority objective of our economic and social policies, to enable all men and women to secure and secure livelihoods through productive employment and work of their own free choice.” Since then, further theoretical and empirical studies on sustainable livelihood have been carried out [39].

On the basis of the concept of sustainable livelihood, some international organizations put forward and gradually improved the sustainable livelihood analysis framework in the 1990s. This framework can help people to understand poverty and can be used to study the livelihoods of farmers. At present, the most widely used sustainable livelihood analysis frameworks include the sustainable livelihood analysis framework that was established by the UK Department for International Development (DFID) in 2000, the household Livelihood Security framework that was proposed by CARE, and the sustainable livelihood approach that was proposed by the United Nations Development Program (UNDP) [40]. Among these, DFID’s sustainable livelihood analysis framework is the most widely used [41].

With the continuous development of research on sustainable livelihood frameworks, many international scholars conducted multi-angle studies in this area. For example, some scholars put forward diversified livelihood analysis frameworks [42,43], and others have studied sustainable livelihood security and the association between rural livelihoods and poverty reduction [44,45,46]. Many domestic scholars used the sustainable livelihood framework to study rural poverty and other issues. For example, researchers adopted the sustainable livelihood framework and used factor analysis and a comprehensive factor score method to evaluate the changes in farmers’ livelihood capital before and after land acquisition in terms of two dimensions: level and structure [47]. An analysis of the livelihood capital of farmers living in ecological immigrant areas was done using the sustainable livelihood framework [48]. The sustainable livelihood analysis method was also used to analyze the ecological compensation policy, residents’ livelihood capital, and the sustainable livelihood ability of key national ecological function areas in Shanxi Province, China [49]. Based on classical analyses of the sustainable livelihood framework, scholars quantified the relationship between household assets and livelihood and investigated the willingness of individuals to undergo a household homestead transfer [50,51]. In another study, the concepts of a local livelihood system, livelihood function, livelihood dependence, and livelihood breakthrough were proposed, and a new analysis framework was constructed to study the withdrawal of farmers from free-range pig farming [52]. Researchers used the entropy-based improved TOPSIS method to quantify the livelihood capital index of farmers and used the difference-in-difference method to analyze the impact of farmland consolidation ownership adjustment on the livelihood capital of farmers [53]. The impact of rural construction on the livelihood sustainability of targeted poverty alleviation of households was also studied from the perspectives of sustainability and vulnerability [54]. Researchers introduced residents’ disaster avoidance preparedness measures into the framework of sustainable livelihood and explored the relationships between residents’ livelihood capital and their willingness to evacuate and relocate [55,56,57].

To sum up, research on the classification of risk shocks and the adverse effects of risk shocks is relatively mature. In addition, a sustainable livelihood has a strong reference value for the prospective and comprehensive judgment of farmers’ welfare level and its future development trend. This concept was widely applied in domestic and foreign livelihood research. Previous studies provide a good theoretical basis for this study, but there are still the following deficiencies in this area of research: First, in terms of research objects, scholars mostly studied the sustainable livelihood of farmers by using survey data from a certain region, and there has been a lack of data collection and collation at the national level for the study of sustainable livelihood achievement by farmers. Second, most previous studies focused on the direct impact of single risk factors on farmers and did not consider the long-term risk impact from the perspective of achieving a sustainable livelihood.

### 2.2. Theoretical Analysis

The UK Department for International Development (DFID) proposed the Sustainable Livelihoods Analysis Framework, which consists of five parts: vulnerability background, livelihood capital, transformation structures and processes, livelihood strategies, and livelihood outcomes (see Figure 1). According to the framework, livelihood risks are present throughout the whole process of achieving a sustainable livelihood, showing that people live in “risk-vulnerable environments,” and risk and vulnerability directly affect people’s livelihood capital, viable capacity, and livelihood strategy choices, and indirectly affect livelihood consequences. Farmers directly face various risk shocks in their livelihood activities, and vulnerable farmers may experience an unexpected income reduction or welfare loss, increasing the possibility of livelihood turmoil and reducing their sustainable livelihood capacity.

Natural disasters are common risks that are faced by farmers. China is prone to natural disasters. Earthquakes, floods, droughts, debris flows, typhoons, and other types of natural disasters are destructive and persistent. Natural disasters can cause loss of life and property and may significantly impact farmers’ livelihoods, seriously restricting the sustainable and healthy development of the rural economy. For vulnerable farmers, household risk resistance is poor. Once agricultural production is subjected to natural disasters, the production of agricultural products will reduce, and the household income of farmers will decline. In addition, it is difficult to achieve risk transfer, meaning that families are easily led into a state of poverty [58,59].

Health risk is also a common risk type encountered by farmers, and it is also the risk type with the biggest impact on farmers’ livelihoods. The term health risk shock refers to the phenomenon whereby, after suffering from a type of health shock, such as a serious or chronic disease, farmers may experience decreased agricultural production efficiency and increased medical costs, increasing their economic vulnerability. Health risks are the root cause of rural households falling into poverty and an important determinant of long-term poverty [60]. Due to poor access to medical facilities in rural areas and the fact that most of the farmers work all year round, farmers are in poor physical condition. At the same time, due to a lack of funds, farmers often do not pay attention to their own health status, resulting in the occurrence of serious diseases, chronic diseases, and other accidents. Health risks have a significant negative impact on farmers, which is mainly manifested in the reduction of their dietary nutrition level in the short term. In addition, these risks may reduce or delay the purchase of daily consumer goods and may even reduce investment in children’s educational capital, thereby reducing the children’s future expected income. In the long run, health risks will reduce the farmer’s working hours, resulting in a decline in the average income of the whole family.

Based on the above analysis, the following research hypothesis was proposed: environmental and health risks will negatively impact the achievement of a sustainable livelihood by farmers.

## 3. Research Design

### 3.1. Data Source

This study used rural household samples from the China Family Panel Studies (CFPS) database collected in 2010, 2012, 2014, 2016, and 2018. This project was initiated by The China Social Science Survey Center of Peking University in 2010. It involved field research in 25 provinces and cities across the country. The collected data cover a wide range of levels and have high reliability. Data from individuals, families, and communities are included, providing a good basis for this research.

In this study, Stata 15.0 was used to process the data. First, household and personal data were processed, and data on core variables, such as livelihood capital, natural, and health risks, were obtained. Then, personal IDs were used as identification codes to match family data with personal data, and individuals with missing values were eliminated. Finally, 3906 farming households and 19,530 valid data samples were obtained. Regarding regional differences in the study area, this study took China’s topographic features as the classification standard to divide the farms into those situated on plains, hills, and mountains.

### 3.2. Sustainable Livelihood Index Selection and Weight Measurement Method

The sustainable livelihood index is based on the identification of important types of capital that are related to livelihood (human capital, natural capital, physical capital, social capital, and financial capital) and the influence of social conditions. It focuses on exploring ways to improve the ability to optimize household capital. Based on the sustainable livelihood analysis framework, this study analyzed five sustainable livelihood index indicators: (1) human capital included the average education level of the labor force, the labor force proportion, and the trained labor force proportion; (2) natural capital was measured using the land area that was owned by farmers; (3) physical capital included the housing area, housing value, and living durable goods; (4) social capital included two indicators: relationships with family and friends and relationships with neighbors; and (5) financial capital included five indicators: wage income, operating income, property income, transfer income, and savings (see Table 1 for details).

Based on research by Wu. et al., this study used the entropy weight method to determine the weight of the index system of the achievement of a sustainable livelihood by farmers. The entropy weight method measures the amount of information provided by each index from a mathematical point of view and determines the weight of each index on this basis. As an objective weighting method, it can reduce the interference from human factors on the evaluation results, scientifically calculate the entropy weight of each index, and produce more scientific evaluation results. The specific measurement steps are as follows:

First, dimensionless processing of the indexes is carried out:(1)$${X^{\prime }}_{ij}=\frac{X_{ij}-\min\left(X_{j}\right)}{\max\left(X_{j}\right)-\min\left(X_{j}\right)}$$

In Formula (1), ${X^{\prime }}_{ij}$ represents the normalized value of index j of sample i. $X_{ij}$ represents the variable value of index j of sample i. $\max\left(X_{j}\right)$ represents the maximum value of index j, and $\min\left(X_{j}\right)$ represents the minimum value of index j.

Second, the information entropy of each index is calculated:(2)$$E_{j}=-\frac{1}{\ln n}\sum_{i=1}^{N}P_{ij}\ln P_{ij}$$

Among them, $P_{ij}=\frac{{X^{\prime }}_{ij}}{\sum_{i=1}^{N}{X^{\prime }}_{ij}}$.

To determine the weight of each index, the entropy value of each index is calculated using Formula (2) (*E*_1_, *E*_2_, …, *E_m_*). To calculate the weight of each index using the entropy value method, the following equation is used:(3)$$W_{j}=\frac{1-E_{j}}{\sum E_{j}}\left(0\leq j\leq m\right)$$

Finally, the sustainable livelihood index is calculated according to the weight of the index:(4)$$Z_{i}=\overset{14}{\underset{i=1}{\sum }}X_{ij}*W_{j}$$

### 3.3. Variables Selected

(1) Explained variables: The explained variable used in this study was the sustainable livelihood index of farmers. The weight of each dimension index was determined using the entropy weight method, as mentioned above, and the sustainable livelihood index of farmers was calculated.

(2) Core explanatory variables: Two core variables were used. First, the environmental risk shock index, which resulted from the complex and varied natural disasters present in the country, including drought, floods, hail, typhoon, storm, pests, earthquakes, landslides, and other disasters, and may lead to other disasters, was used. This study used farmers suffering from natural disasters to measure the environmental risk shock indicator. The second factor was the health risk impact index, which can be measured using a variety of methods. In this study, based on the research by Chu. et al. [61], risk shock from major diseases and chronic diseases was selected to determine whether farmers suffered from this type of shock. Specifically, the measure of serious disease risk shock was determined using the proportion of hospitalized adults in the family compared with the total number of adults in the family. The chronic disease risk shock was measured as the proportion of adults with chronic disease from the total number of adults in the family.

(3) Control variables: The control variables that were introduced in this study included household characteristics, household head characteristics, and village-level characteristics. The characteristics of the head of household included their age, sex, education level, and party member status. Family characteristics included the family size, an adult health self-assessment, essential living expenses, adult medical insurance purchase, government assistance, and medical expenses; village-level characteristics included mineral resources, collective enterprises, and tourism resources owned by each village (see Table 2 for details).

### 3.4. Model Selected

Taking the sustainable livelihood index measured using the above method as the explained variable; the environmental risk, major disease risk, and chronic disease risk as explanatory variables; and the household characteristics, household head characteristics, and community characteristics as control variables, the Tobit regression model was constructed as follows:(5)$$Y=a+bX_{i,t}+cControl_{i,t}+\varepsilon_{i,t}$$

In the above equation, Y represents the sustainable livelihood index of farmers, *X* represents the risk shocks suffered by farmers, *i* represents the types of risk shock, including environmental risk shock, serious disease risk shock, and chronic disease risk shock; *t* represents the year; control represents a series of control variables; a and $\varepsilon_{i,t}$ represent a constant term and a random disturbance term, respectively; and b and c represent coefficients to be determined.

## 4. Results and Discussion

### 4.1. Descriptive Statistical Characteristics

Table 3 shows the descriptive statistical characteristics of the variables. The maximum value of the sustainable livelihood index was 1.6173, the minimum value was 0.0193, and the mean value was 0.2527, indicating that there was a large gap in the ability of farmers to achieve a sustainable livelihood ability. The ability of farmers to achieve a sustainable livelihood was found to be at a low level. The maximum value of the environmental risk shock was 8 and the minimum value was 0, indicating that environmental risk types in China were complex and diverse. The minimum value of the major disease risk shock and the maximum value of chronic disease risk shock were 0 and 1, and the mean values were 0.1019 and 0.1533, respectively, indicating that the proportions of adults in peasant households suffering from serious disease risk and chronic disease risk were small. The maximum self-assessment value was 5, the minimum value was 0, and the mean value was 3.0874, indicating that most adult farmers were healthy. The maximum value for the government help variable was 1, and the average value was 0.5758, indicating that more than half of the farmers had accepted government help.

### 4.2. Model Regression Results

Table 4 shows the regression results for the impact of risk shocks on farmers’ achievement of a sustainable livelihood. Column (1) presents the regression results for the full sample. The environmental risk of the core explanatory variable was found to be significantly negative at the 1% level, the major disease risk was significantly negative at the 10% level, and the chronic disease risk was significantly negative at the 5% level. Therefore, environmental risk shocks and health risk shocks were found to have significant negative impacts on the achievement of a sustainable livelihood by farmers. Column (2) presents the regression results of the sample from the plain area. The environmental risk of the core explanatory variable was found to be significantly negative at the 5% level, and the risks from major diseases and chronic diseases were significantly negative at the 1% level. Therefore, the environmental and health risks were important factors that affected the achievement of a sustainable livelihood by farmers in plain areas. Column (3) reports the regression results of samples from mountainous areas. The core explanatory variable environmental risk was found to be significantly negative at the 1% level. Therefore, it can be seen that farmers that were living in mountainous areas were the most vulnerable to environmental risk, and this type of risk was the most important factor that affected the achievement of a sustainable livelihood by mountainous area farmers. In addition, the risk of chronic disease was found to be significantly negative at the 10% level. Major disease risk was also found to have a negative impact on the achievement of a sustainable livelihood by rural households in mountainous areas, but the effect was not significant. The reason for this may have been that the accessibility of rural medical facilities in mountainous areas is poor. In addition, due to constraints from economic conditions, most rural households do not pay attention to their own health statuses and often do not use formal medical care. Column (4) shows the regression results of samples from hilly areas. Chronic disease risk was found to have a negative impact on the achievement of a sustainable livelihood by farmers at the 5% level, while a significant difference between environmental risk and major disease risk was found.

### 4.3. Endogeneity Test

Theoretically, there may be endogeneity problems that are caused by the causal relationship between chronic diseases diagnosed by peasant households and the sustainable livelihood index of peasant households. In this study, the regression results verified that farmers’ ability to achieve a sustainable livelihood decreased after exposure to chronic disease risk. With the loss of their ability to achieve a sustainable livelihood, farmers may maintain their long-term welfare by reducing the nutritional level of their diet or by reducing or delaying the purchase of consumer goods. This increases the risk of the farmer being diagnosed with a chronic disease. In order to verify the endogeneity of the variables, two variables were selected as instrumental variables: “the distance of the family to the nearest hospital/medical point” and “the time required for the family to travel to the nearest city (town) commercial center.”

Table 5 shows the endogeneity regression results. According to the results of the first stage of regression, the regression estimation coefficients of instrumental variables on the endogenous variables passed the significance test at the 1 and 10% levels, indicating that the selected instrumental variable was valid. In addition, the F-statistic was 239.32, which is greater than the assumption of 10; therefore, there was no weak instrumental variable problem. The regression results of the second stage showed that the Prob > chi^2^ of the Wald test was 0.0000, indicating that there was no serious endogeneity problem [62,63]. The proportion of the diagnosed chronic diseases was found to be negatively correlated with the sustainable livelihood index of the farmers at the 1% level, which verified the conclusion of this study.

### 4.4. Robustness Test

In order to test the robustness of the previous estimation results, this study re-estimated the parameters by adopting different sustainable livelihood indexes and eliminating some variables. Using the method developed by Sun. et al., this study included the children’s education in the evaluation of sustainable livelihood and optimized the traditional framework of sustainable livelihoods to reflect the sustainability of intergenerational farmers. Two variables were added into the original sustainable livelihood index system to calculate the sustainable livelihoods index value: farmers’ education expenses and school attendance by school-age children. In addition, four variables—government assistance, medical insurance for farmers, the educational background of the head of the household, and the party membership status of the head of the household—were excluded, and then the regression was carried out.

Table 6 reports the robustness test results. The first column presents the regression results for the improved sustainable livelihood index. A significant negative correlation between environmental risk shock and the sustainable livelihood index of farmers was found at the 1% level. Major disease and chronic disease risks were found to have significant negative impacts on the sustainable livelihood index of farmers at the 5% level. The second column presents the regression results after excluding some variables. Both environmental risk and chronic disease risk were found to have significant negative impacts on the sustainable livelihood index of farmers at the 1% level. A significant negative correlation between the risk shock of serious diseases and the sustainable livelihood index of farmers at the 10% level was found. Therefore, the estimation results in this study were robust.

## 5. Discussion

China has entered the “post-poverty alleviation” era, where the achievement of sustainable livelihoods by farmers has become a focus. The impact of risk shock on farmers’ livelihoods has long been a focus of academic attention. Different from the existing literature, this study selected five years of data from 3906 peasant households from the CFPS database. Based on the DFID sustainable livelihood framework, a sustainable livelihood index system for farmers was built. The entropy weight method was used to measure the weight of each sustainable livelihood index and the sustainable livelihood index of farmers was calculated. The Tobit model was used to explore the impacts of different types of risk shocks on farmers’ ability to achieve a sustainable livelihood from the perspective of risk shocks.

The results showed that environmental risk shock, major disease risk shock, and chronic disease risk shock all had significant negative effects on the achievement of a sustainable livelihood by farmers. This is in accordance with previous research showing that risk has adverse effects on farmers. In order to better reflect the differences in research areas, this study took the topographic characteristics of China as the classification standard and divided the samples into plains, hills, and mountains. The results showed that environmental risk, major disease risk, and chronic disease risk had significant negative impacts on the achievement of a sustainable livelihood by farmers living in plain areas. Environmental risk was found to have a significant negative impact on the achievement of a sustainable livelihood by farmers living in mountainous areas. Thus, farmers living in mountainous areas were the most vulnerable to environmental risk, and this was also the most important factor that affected the achievement of a sustainable livelihood by farmers living in mountainous areas. In addition, the risk from serious diseases also had a negative impact on the achievement of a sustainable livelihood by rural households in mountainous areas (though no significant difference was found), possibly due to the poor access to medical facilities in rural mountainous areas and the limitations from economic conditions, as these factors mean that most rural households do not pay attention to their own health conditions and often do not use formal medical care. Chronic disease risk was found to have a negative impact on the achievement of a sustainable livelihood by farmers in hilly areas, but no significant impacts of environmental risk and serious disease risk were identified.

In addition, the possible endogeneity problems in this research were tested, and the results showed that there were no serious endogeneity problems in this study. On this basis, the robustness test was carried out by improving the sustainable livelihoods index and removing some variables. The results showed that both environmental risks and health risks had significant negative impacts on the achievement of sustainable livelihoods by farmers, which confirmed the research results of this study.

## 6. Conclusions

The results of this study were as follows: (1) There were large gaps in the sustainable livelihood capacities of farmers, and the sustainable livelihood capacity of most farmers was still at a low level. (2) Environmental risk shock, major disease risk, and chronic disease risk all had significant negative effects on farmers’ ability to achieve a sustainable livelihood, and environmental risk and health risk were important adverse factors that affected farmers’ ability to achieve a sustainable livelihood. (3) Environmental risk shock, serious disease risk, and chronic disease risk all had significant negative impacts on farmers’ achievement of a sustainable livelihood in plain areas, but the impacts of serious disease risk and chronic disease risk on farmers’ achievement of a sustainable livelihood were stronger than that of environmental risk shock. (4) The types of environmental risk in China were complex and diverse, and environmental risk was the most important factor that affected the achievement of a sustainable livelihood by rural households in mountainous areas. Risk shock from chronic diseases was also an important adverse factor that affected the achievement of a sustainable livelihood by rural households in mountainous areas.

Based on the above conclusions, the following policy recommendations are proposed. First, the risk-coping ability of farmers should be improved, the relevant education of farmers’ risk resistance should be strengthened, the identification and cognition of farmers’ risk types should be enhanced, and the most effective risk prevention behavior strategy should be used to reduce the risk from different categories. Second, we should enhance farmers’ awareness of risk management and prevention, broaden the channels for farmers to obtain information, guide farmers to pay attention to information in a timely manner, and encourage farmers to buy appropriate agricultural insurance to reduce the impact of natural disasters. Third, the construction of rural medical infrastructure and a medical insurance system should be improved to provide farmers with better medical services and to encourage them to use formal medical services.

## Figures and Tables

**Figure 1 ijerph-18-10955-f001:**
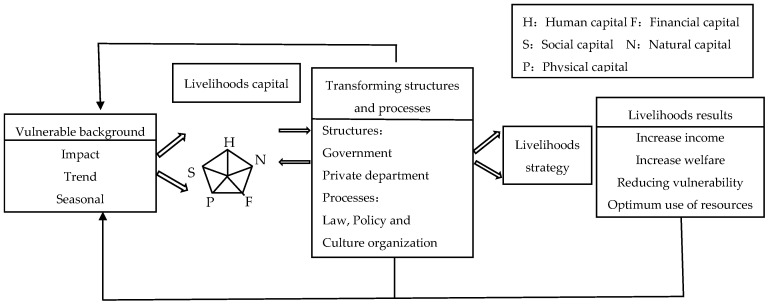
Sustainable Livelihoods Framework.

**Table 1 ijerph-18-10955-t001:** Sustainable livelihood indicator system and weights.

Category	Variable Name	Variable Definitions	Weight
>Human capital	The average level of education in the workforce	1 = Cannot read, 2 = primary school, 3 = junior high school, 4 = high school, 5 = college degree or above	0.0323
	Labor proportion	The proportion of household labor force in the total household population	0.0320
	Proportion of the trained workforce	The proportion of the workforce trained in professional skills of the total workforce	0.0257
Natural capital	Land area	Family per capita land area (mu)	0.0471
Physical capital	Housing area	Per capita living area of a family (square meters)	0.0410
	Housing value	Current market value of the family house (ten thousand CNY, logarithm)	0.0276
	Durable goods	Value of durable goods (CNY, logarithm)	0.0260
Social capital	Relationships with friends and relatives	Contact with relatives and friends	0.0276
	Relationships with neighbors	Number of contacts with neighbors	0.0262
Financial capital	Wage income	Element (logarithm)	0.0414
	Operating income	Element (logarithm)	0.0258
	Property income	Element (logarithm)	0.0273
	Transfer income	Element (logarithm)	0.0171
	Savings	Element (logarithm)	0.0933

**Table 2 ijerph-18-10955-t002:** Variable definition table.

Category		Variable Name	Variable Definitions
Explained variable		Sustainable livelihood index	Sustainable livelihoods index score
Explanatory variables		Environmental risk shock	The extent to which families were exposed to natural disasters
		Major disease risk shock	The proportion of adults in the household who were hospitalized as a percentage of the total number of adults in the household.
		Chronic disease risk shock	The proportion of adults with chronic diseases out of the total number of adults in the family
Control variables	Family characteristics	Family size	Total family size
		Self-evaluation of health	Mean values of self-reported adult health at home
		Medical expenses	Family health care expenditure (CNY, logarithm)
		Living expenses	Necessary expenses for family life (CNY, logarithm)
		Government help	Whether they received government subsidies (1 = yes, 0 = no)
	Household head characteristics	Age	The age of the head of the household
		Gender	Gender of the head of the household (1 = male, 0 = female)
		Education level	Highest degree completed by the head of the household (1 = junior high school and above, 0 = primary school and below)
		Party membership	Is the head of the household a party member? (1 = yes, 0 = no)
	Village characteristics	Mineral resources	Whether the village has mineral resources (1 = yes, 0 = no)
		Collective enterprise	Whether the village has a collective enterprise (1 = yes, 0 = no)
		Tourism resources	Is the village a tourist village (1 = yes, 0 = no)

**Table 3 ijerph-18-10955-t003:** Descriptive statistical characteristics of the variables.

	Sample Size	Mean	Standard Deviation	Minimum Value	Maximum Value
Sustainable livelihood index	19,530	0.2527	0.1130	0.0193	1.6173
Environmental risk shock	19,530	2.0320	1.7231	0.0000	8.0000
Major disease risk shock	19,530	0.1019	0.2155	0.0000	1.0000
Chronic disease risk shock	19,530	0.1533	0.2650	0.0000	1.0000
Family size	19,530	4.1465	1.8905	1.0000	16.0000
Self-evaluation of health	19,530	3.0874	1.0764	0.0000	5.0000
Medical expenses	19,530	6.7608	2.5692	0.0000	13.5144
Living expenses	19,530	9.4853	1.0977	0.0000	17.7000
Government help	19,530	0.5758	0.4942	0.0000	1.0000
Medical insurance	19,530	0.8832	0.2510	0.0000	1.0000
Householder’s age	19,530	47.9200	17.4177	0.0000	94.0000
Householder’s gender	19,530	0.5651	0.4958	0.0000	1.0000
Householder’s education level	19,530	0.2325	0.4224	0.0000	1.0000
Householder’s party membership	19,530	0.0798	0.2710	0.0000	1.0000
Mineral resources	19,530	0.0947	0.2928	0.0000	1.0000
Collective enterprise	19,530	0.0289	0.1676	0.0000	1.0000
Tourism resources	19,530	0.0148	0.1210	0.0000	1.0000

**Table 4 ijerph-18-10955-t004:** Model estimation results.

Variable Name	(1) Entire Sample	(2) Plains	(3) Mountains	(4) Hills
Environmental risk shock	−0.0029 *** (−3.39)	−0.0045 ** (−2.52)	−0.0057 *** (−2.87)	0.0016 (1.53)
Major disease risk shock	−0.0052 * (−1.72)	−0.0154 *** (−2.73)	−0.0037 (−0.53)	−0.0035 (−0.85)
Chronic disease risk shock	−0.0043 ** (−2.10)	−0.0113 *** (−3.00)	−0.0115 * (−1.92)	−0.0075 ** (−2.14)
Family size	−0.0109 *** (−21.21)	−0.0066 *** (−7.72)	−0.0113 *** (−9.86)	−0.0076 *** (−10.82)
Self-evaluation of health	−0.0023 *** (−3.44)	−0.0017 (−1.34)	−0.0027 * (−1.68)	−0.0066 *** (−7.34)
Medical expenses	0.0003 (1.06)	0.0017 *** (3.71)	0.0010 * (1.75)	0.0013 *** (3.58)
Living expenses	0.0140 *** (21.42)	0.0045 *** (6.50)	0.0097 *** (6.39)	—
Government help	0.0338 *** (24.83)	0.0348 *** (14.25)	0.0383 *** (12.36)	—
Medical insurance	0.0003 (0.11)	−0.0002 (−0.06)	−0.0028 (0.48)	0.0171 *** (5.16)
Householder’s age	0.0003 *** (6.97)	0.0003 *** (3.12)	0.0002 ** (2.17)	0.0004 *** (5.79)
Householder’s gender	0.0065 *** (4.80)	0.0062 *** (2.58)	0.0036 (1.18)	0.0068 *** (3.67)
Householder’s education level	−0.0066 *** (−3.84)	−0.0085 *** (−2.81)	0.0018 (0.48)	−0.0055 ** (−2.29)
Householder’s party membership	0.0033 (1.26)	0.0023 (0.49)	0.0005 (0.95)	0.0036 (0.98)
Mineral resources	−0.0102 * (−1.94)	0.0310 (0.49)	−0.0010 (−0.07)	−0.0155 *** (−2.69)
Collective enterprise	−0.0025 (−0.2)	—	−0.0032 (−0.14)	0.03548 *** (2.67)
Tourism resources	0.0083 (0.92)	0.0113 (0.50)	0.0049 (3.35)	−0.03506 *** (−3.53)
Observations	19,530	6530	3285	6948

Note: ***, ** and * denote statistical significance at the 1, 5, and 10% levels, respectively. The values in parentheses are the Z-statistics.

**Table 5 ijerph-18-10955-t005:** Endogeneity test results.

	First-Stage Regression Results	Second-Stage Regression Results
	Proportion of People Diagnosed with Chronic Diseases	Sustainable Livelihood Index for Farmers
Distance from home to nearest hospital/medical point	−0.0027 *** (0.0005)	—
Time from home to the nearest city (town) business center	−0.0001 * (0.00002)	—
Proportion of people diagnosed with chronic disease	—	−0.2717 *** (0.0516)
F-statistic	239.32	—
Control variables	Yes	Yes
	Prob > F = 0.0000	Prob > chi^2^ = 0.0000

Note: *** and * denote statistical significance at the 1 and 10% levels, respectively. The values in parentheses are the standard errors.

**Table 6 ijerph-18-10955-t006:** Robustness test results.

Variable Name	Optimized Sustainable Livelihood Index	Sustainable Livelihood Index
Environmental risk shock	−0.0031 *** (−3.61)	−0.0030 *** (−3.80)
Major disease risk shock	−0.0067 ** (−2.16)	−0.0060 * (−1.92)
Chronic disease risk shock	−0.0054 ** (−2.11)	−0.0093 *** (−3.63)
Family size	−0.0075 *** (−14.43)	−0.0106 *** (−20.20)
Self-evaluation of health	−0.0018 ** (−2.56)	−0.0060 *** (−9.09)
Medical expenses	0.0004 (1.43)	0.0006 ** (2.12)
Living expenses	0.0157 *** (23.86)	0.0180 *** (27.07)
Government help	0.0347 *** (25.33)	—
Medical insurance	−0.0011 (−0.45)	—
Householder’s age	0.0003 *** (5.52)	0.0005 *** (11.37)
Householder’s gender	0.0064 *** (4.71)	0.0071 *** (5.45)
Householder’s education level	−0.0061 *** (−3.52)	—
Householder’s party membership	0.0021 (0.79)	—
Mineral resources	−0.0103 * (−1.95)	−0.0085 * (−1.80)
Collective enterprise	−0.0007 (−0.05)	−0.0116 (−1.02)
Tourism resources	0.0037 (0.41)	−0.0024 (−0.29)
Observations	19,530	18,875

Note: ***, **, and * denote statistical significance at the 1, 5, and 10% levels, respectively. The values in parentheses are the Z-statistics.

## Data Availability

The data are available online at http://www.isss.pku.edu.cn/cfps, (accessed on 10 October 2020).

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
