# Peer review of "Impact of Environmental and Health Risks on Rural Households’ Sustainable Livelihoods: Evidence from China"

_ijerph, 2021, doi:10.3390/ijerph182010955_

Round 1

Reviewer 1 Report

The exporuse to environmental risks and the distribution of these risks between different stakeholder groups are important social questions. The paper contributes to the discussion by analysing the stakeholder group of farmes with respect to environmental risks. 

The paper is well written and the methods clearly described. 

Author Response

Dear reviewer and editor,

Thank you very much for your affirmation of the article. And we have carefully revised the following according.

Firstly, we have checked and revised the article for spelling and grammar.

Secondly, we modified the introduction. The specific modifications are as follows: (1) we added reference to the section of the improvements experienced by China in the fight against poverty. (2) we added the more accurate and detailed description of the subject and research objectives. (3) The research structure of the paper was added in the last part of the introduction.

Thirdly, We revised the literature review as follows:(1) We added the researches published in or about China. (2) The format of literature review has been modified.

In addition, the definition of risk shock considered in this study was supplemented in the theoretical analysis.

Finally,We added a discussion section to discuss the results of the study in detail.

The above is the revision content corresponding. If there are deficiencies, please kindly comment from the experts. We will make serious modifications according to the requirements of the experts.

Reviewer 2 Report

 1. The sustainable livelihood of peasant  households in China as a research topic is important from the scientific and practical side.

2. The concentration of the study on environmental and health risks as a main factors of sustainable livelihood is justified.

3. Introduction to the study, literature review and theoretical analysis are appropriate. Distinguishing the five parts ofsustainable livelihood framework is acceptable.

4. Data sources and methods used are adequate. Variable used for analysis may be discutable, however they are generally suitable for Chinese conditions.

5. Statistical model used and test results for mode estimations are correct.

6. Four points of concentration distinguished in conclusions concerning risk shocks influencing the farmers sustainable livelihood  well reflects research results . Also,proposed three types of policy recommendations well summarise the outcomes of the paper.

Author Response

(The authors gave the same response as above.)

Reviewer 3 Report

This article analyses and discusses different types of risk impact influence on the sustainable livelihood of Chinese farmers using a huge amount of data from the China Family Panel Studies (CFPS) database. In this sense, it is important to point out that the quantitative information used constitutes an adequate sample to support the results of the empirical work and, therefore, to validate the conclusions reached by the authors based on these results. The method used is also adequate, in the opinion of the reviewer, and although it can be explained and exposed in a somewhat clearer way for the reader, its presentation is acceptable. For all the above, the reviewer considers that the research does not present substantive problems, although some formal aspects can be mentioned that must necessarily be corrected.

Firstly, grammatical and even spelling errors have been found in some parts of the text, including in the abstract, something that obviously has to be corrected. Second, the statements that the authors make in the introduction, regarding the improvements experienced by China in the fight against poverty, are not supported by any reference, so the reader cannot get an idea of ​​the extent to which such improvements have occurred. Authors must add some references in this case. The introductory section, in fact, should be rewritten: (1) ... to present the investigated topic in a better way; (2) ... to collect the research objectives more precisely, since in the current text these are expressed in a vague and disorderly way; (3) ... to offer a somewhat greater detail of the source used, and the virtues and problems that its use presents; and (4) to set the structure of the research, something that is necessary to help the reader keep in mind the points that make up the structure of the paper.

In other order, it may also be interesting, especially for readers not familiar with the studied subject, to make a brief definition or description of the risks analyzed and considered in the paper. Some of them are obvious, but others would require a brief explanation of what they are, including examples if necessary.

Regarding the bibliographic review that is carried out, the reviewer understands that it can be improved. On the one hand, there is an extensive international bibliography on the subject, and, however, the authors hardly use research published in or about China. We recommend authors to make an effort in this regard, as this, in addition to improving the research framework, will give their work a more international character. On the other hand, the bibliographic review carried out is unstructured for a long time and, in its current format, it does not constitute a good framing of the research, in the opinion of the reviewer, among other things, because it does not adequately support either the method or the approach used in the research.

Either way, the formal suggestions for improvement that are made do not cloud the research results, so the work could be published as long as these minor changes are addressed.

Author Response

Dear reviewer and editor,

 Thank you very much for your valuable advice. Your opinions are very important. We have carefully revised the following according to the review opinions of the paper.

Firstly, we have checked and revised the article for spelling and grammar.

Secondly, we modified the introduction. The specific modifications are as follows:

(1) we added reference to the section of the improvements experienced by China in the fight against poverty. (references: 1. X, Wang.; X, Zhang. An Explanation of China’s Experience in Eliminating Absolute Poverty and the Orientation of Relative Poverty Governance in the Post-2020 Era. Chinese Rural Economy. 2021, 02, 2-18. 2. S, Wang.; M, Liu. From Absolute Poverty to Relative Poverty: Theoretical Relationships, Strategic Shifts and Policy Priorities. Journal of South China Normal University (Social Science Edition). 2020, 06, 18-29+189. 3. J, Sun.; Q, Zhang. Research on West Development and the Expansion of the Opening up of the Western Region. Journal of Xinjiang Normal University (Edition of Philosophy and Social Sciences). 2021, 42, 04, 79-91+2.)

(2) we added the more accurate and detailed description of the subject and research objectives. The additions are as follows:“This article was based on analysis of sustainable livelihoods framework and the China Family Panel Studies (CFPS) database, built an index system of sustain-able livelihoods of farmers, measured the weight of sustainable livelihood of target indexes by the entropy weight method and measured sustainable livelihood index of farmers, used the Tobit model to discuss the impact of different types of risks on the sustainable livelihood of farmers. On the basis, this article tried to explain the negative impact of risk shock on farmers' sustainable livelihood from the perspective of microcosmic level of farmers in different regions, and according to different agricultural risk categories, developed effective risk prevention and response strategy, provided decision-making idea for farmers to raise the lev-el of sustainable livelihoods., It has certain practical reference value.”

(3) The research structure of the paper was added in the last part of the introduction. The additions are as follows:“The rest of this paper was arranged as follows: The second part is literature re-view and relevant theoretical analysis; The third part introduces the sources of data used in this paper, the selection and weight measurement methods of sustainable livelihood indicators, variables and models adopted; The fourth part is the main part of the paper, which is the correlation analysis of the impact of environmental risk and health risk on the sustainable livelihoods of farmers, as well as the endogenous problem test and robustness test; The fifth part is the discussion of the results, and the sixth part is the conclusion and relevant policy suggestions.”

Thirdly, We revised the literature review as follows:(1) We added the researches published in or about China. The addition is as follows: “And some researchers had introduced residents' disaster avoidance preparedness measures into the framework of sustainable livelihood, and explored the relationships between residents' livelihood capital and their evacuation and relocation willingness” (reference: W, Zhou.; Z, Ma.; S, Guo.; X, Deng.; D, Xu. Livelihood capital, evacuation and relocation willingness of residents in earthquake stricken areas of rural China. Safety science. 2021, 141.). (2) The format of literature review has been modified.

Finally, the definition of risk shock considered in this study was supplemented in the theoretical analysis. The increase is as follows:“Health risk shock refers that after suffering from health shock such as serious disease or chronic disease, farmers maybe have the decrease of efficiency of agricultural production and increase of medical cost, thus increasing their economic vulnerability.”

The above is the revision content corresponding to the review opinions put forward by the review experts. If there are still deficiencies, please kindly comment from the experts. We will make serious modifications according to the requirements of the experts.

Reviewer 4 Report

Dear Authors,

I think this manuscript, "Impact of Environmental and Health Risks on Rural Households' Sustainable Livelihoods: Evidence from China," has been duly written and prepared to get published in the journal IJERPH. I suggest this manuscript has been almost close and ready for official publication. I recommend adding short paragraphs of discussions after the results. The discussions should be based on the results. In the present format, the results address discussion as well as results. Our potential readers will enjoy the manuscript in the organizing text format by two separated sections of discussion and result.

Author Response

Dear reviewer and editor,

Thank you very much for your valuable advice. Your opinions are very important. We have carefully revised the following according to the review opinions of the paper.

We added a discussion section to discuss the results of the study in detail. The limitations increase is as follows:

China has entered the "post-poverty alleviation" era, and the sustainable livelihoods of farmers has become one of the focus issues in the "post-poverty alleviation" era. The impact of risk shock on farmers' livelihoods has long been the focus of academic attention. Different from the existing literature, this article selected 3906 peasant households in the five years tracking data from CFPS database, based on analysis of DFID sustainable livelihood framework, built the index system of sustainable livelihoods of farmers. The entropy weight method was used to measure the weight of each sustainable livelihood index and calculate the sustainable livelihood index of farmers. On this basis, the Tobit model was used to explore the impact of different types of risk shocks on farmers' sustainable livelihoods from the perspective of risk shocks.

The results showed that environmental risk shock, major disease risk shock and chronic disease risk shock all have significant negative effects on the sustainable livelihoods of farmers. This is the same as scholars' research conclusion that risk will bring adverse effects on farmers. In order to better reflect the differences of research areas, this paper took the topographic characteristics of China as the classification standard, and divided the samples into plains, hills and mountains. The results showed that environmental risk, major disease risk and chronic disease risk have a significant negative impact on the sustainable livelihoods of farmers in the plain area. Environmental risk has a significant negative impact on the sustainable livelihood of farmers in mountainous areas, so it can be seen that environmental risk is the most vulnerable type of risk for farmers in mountainous areas, and also the most important factor affecting the sustainable livelihoods of farmers in mountainous areas. In addition, the risk of serious diseases will also have a negative impact on the sustainable livelihoods of rural households in mountainous areas, but there is no significant difference, which may be due to the poor accessibility of medical facilities in rural areas in mountainous areas, and due to the limitations of their own economic conditions, most rural house-holds do not pay attention to their own health conditions and often do not use formal medical care. The impact of chronic disease risk has a negative impact on the sustainable livelihood of farmers in hilly areas, while there is no significant impact of environmental risk and serious disease risk.

In addition, the possible endogeneity problems in this research were tested and the results showed that there were no serious endogeneity problems in this study. On this basis, the robustness test was carried out by means of improving the sustainable livelihoods index and removing some variables. The results showed that both environmental risks and health risks have a significant negative impact on the sustainable livelihoods of farmers, which confirms the research results of this paper.

The above is the revision content corresponding to the review opinions put forward by the review experts. If there are still deficiencies, please kindly comment from the experts. We will make serious modifications according to the requirements of the experts.
